# A New Laboratory Scale Olive Oil Extraction Method with Comparative Characterization of Phenolic and Fatty Acid Composition

**DOI:** 10.3390/foods12020380

**Published:** 2023-01-13

**Authors:** Miguel D. Ferro, Maria João Cabrita, José M. Herrera, Maria F. Duarte

**Affiliations:** 1Alentejo Biotechnology Center for Agriculture and Agro-Food (CEBAL)/Instituto Politécnico de Beja (IPBeja), 7801-908 Beja, Portugal; 2MED—Mediterranean Institute for Agriculture, Environment and Development & CHANGE—Global Change and Sustainability Institute, CEBAL, 7801-908 Beja, Portugal; 3MED—Mediterranean Institute for Agriculture, Environment and Development & CHANGE—Global Change and Sustainability Institute, Departamento de Fitotecnia, Escola de Ciências e Tecnologia, Universidade de Évora, Pólo da Mitra, Ap. 94, 7006-554 Évora, Portugal; 4MED—Mediterranean Institute for Agriculture, Environment and Development & CHANGE, University of Évora, Casa Cordovil, 2nd Floor, R. Dom Augusto Eduardo Nunes 7, 7000-651 Évora, Portugal

**Keywords:** olive oil, extractability, extraction yield, quality, phenolic compounds, fatty acids

## Abstract

The establishment of operation protocols for olive oil (OO) extraction at non-industrial scale is crucial for research purposes. Thus, the present study proposes a simple and cost-effective method for OO extraction at the laboratory scale (LS) level. To validate the proposed methodology, industrial OO extraction (IS) was performed in parallel, using the same cultivars ‘Galega vulgar’ (GV), ‘Cobrançosa’ (COB) and ‘Arbequina’ (ARB) collected from the same orchards, within the same period. Obtained results showed highest extractability for COB and ARB, of about 53%, while GAL showed 50%. All produced OO showed values lower than the regulated limits for the physicochemical parameters (acidity, K232, K268 and ΔK), classifying them as extra virgin OO (EVOO). Highest total phenolic content was observed for COB, with no significant differences (*p*-value > 0.05) between extraction methods. Regarding fatty acid composition, oleic acid (C18:1) showed the lowest percentage for ARB, with about 66% and 68%, for LS and IS, respectively, and the highest for GV with about 72% for both LS and IS. Furthermore, all samples from both extraction methods were compared to the European Community Regulation, with fatty acid composition within the regulated levels for EVOO. This work showed promising results regarding extraction yields and OO extractability, as well as its quality parameters.

## 1. Introduction

Olive oil (OO) is one of the most popular vegetable-based fat sources in recent years, mainly due to its well-known health-promoting properties [1]. In the olive fruit, oil production takes place in the mesocarp cells and is stored in lipo vacuoles. In olive drupes from the north hemisphere, oil accumulation starts to occur by the beginning of July (5 to 7 weeks after full bloom) and finishes by the end of October [2]. The final ripening stage corresponds mainly to water accumulation and, therefore, fresh weight increment. OO is different from other vegetable oils, not only by its chemical characteristics but also due to its extraction process, which is possible only by physical means. Indeed, by definition, virgin olive oil (VOO) is the obtained product extracted from the olive fruits solely by physical means, which do not lead to alterations in the OO, without any further treatment other than washing, decantation, centrifugation and filtration [3].

For OO mechanical extraction there are currently two methodologies: discontinuous-type systems and continuous-type systems. The first ones are those mainly comprised of a millstone combined with hydraulic presses, with a low working capacity and, unavoidably, high labor demand [4], nowadays being viewed as obsolete and superseded by the continuous-type systems. The second type of systems, generally consisting of a mechanical crusher, a malaxer and a decanter, are designated “continuous” as the system operates continuously within the crusher and the decanter, the malaxer operates in batches and is placed between these two continuous components [5]. Within the continuous system, the OO extraction method consists of three main processes: (i) crushing, (ii) malaxation and (iii) centrifugation [6]. After washing, olive fruits are crushed with the sole purpose of facilitating oil droplet release, which will then form larger drops by the malaxation process. Traditionally, the malaxation phase consists in a slow mixing of the olive paste at temperatures ranging from 27 to 32 °C, for 30 to 60 min, depending on the raw material [7]. As referred, the malaxation aims to promote the coalescence of the tiny oil drops generated in the crushing process into larger drops, which can be more easily separated. Crushing and malaxation highly contribute to OO yield, as well as its final properties [8], influenced by the physical, chemical and biochemical reactions occurring during the extraction process [9]. The optimization of OO extraction procedure enhances the activity of enzymes naturally present in olive tissues [8], such as β-glucosidase, leading to desirable reactions directly correlated to the final OO quality. In fact, enzyme activity may be regulated in part by the extraction equipment, as well as by some extraction factors, namely, oxygen availability and, most importantly, temperature [10]. The establishment of operation protocols for OO extraction at non-industrial scale is crucial for research purposes, namely, for comparative studies regarding different orchards’ agronomic practices and their final impact on OO chemical profiles, as well as for olive cultivars assessment and their production yield and OO quality, among many others. This great need for the development of laboratory-scale mills was suppressed, in part, by the appearance of the Abencor system, which facilitated the development of worldwide research into the effects of various agronomic practices on quality indices of OO [11,12,13,14] and OO production yield [15]. The main advantages of the Abencor system reside in its batch mode of operation with great control of processing parameters, enabling the production of reliable results with reduced amount of fruit [16,17].

Nevertheless, and despite this equipment being developed and introduced into the industry research several decades ago, they are still extremely expensive to purchase for the development of laboratory scale research. Therefore, there is still a great need for cost-effective OO extraction methods for pure research purposes, allowing the assessment of agronomic factors’ impacts on OO chemical quality. Thus, the present study proposes a simple and cost-effective method for OO extraction at the laboratory scale, using common laboratory equipment complemented with a commercially available food processor. The obtained OO was further characterized in terms of its physical properties and chemical profiles (phenolic and fatty acids composition). To further validate the purposed methodology, industrial OO extraction was performed in parallel, using the same cultivars—‘Galega vulgar’ (GV), ‘Cobrançosa’ (COB) and ‘Arbequina’ (ARB)—collected from the same orchards, within the same period.

## 2. Materials and Methods

### 2.1. Chemicals and Reagents

All reagents were used as received. Iso-octane was acquired from Sigma-Aldrich (St. Louis, MO, USA) and Folin–Ciocalteu’s reagent from VWR International (Rosny-sous-Bois, Paris, France). Standard gallic acid compound was acquired from Sigma-Aldrich (St. Louis, MO, USA). Double-deionized water was obtained with a Milli-Q water purification system (Millipore, Bedford, MA, USA). Potassium hydroxide pellets were supplied by Merck (Darmstadt, Germany) and *n*-heptane and methanol were purchased from VWR International (Fontenay-sous-Bois, Paris, France). Supelco 37-Component FAME mix, used as a standard solution for identification of FA-components, was purchased from Sigma-Aldrich (Bellefonte, PA, USA). H and air for GC were supplied by Nippon Gases Portugal.

### 2.2. Sampling Collection

Olive fruit sampling was performed at a commercially managed olive orchard at Herdade do Esporão, Reguengos de Monsaraz (38°22′48.1″ N, 7°33′38.4″ W). Sampling was comprised of the cultivars GV, COB and ARB, which were randomly handpicked on 10 October 2021. For each cultivar, the mass of olive fruits collected was as follows: GV—4379.4 g; COB—4434.8 g and ARB—3076.3 g.

### 2.3. Basic Physical Characterizations of Olive Fruit Samples

Both fruit weight (FW) and stone weight (SW) were measured by calculating the average weight of 20 randomly selected olives and their respective stones. The mass of fruit pulp (FP) was obtained by subtracting the SW from the FW value, and fruit pulp to stone ratio (FP/S) by dividing the pulp mass by the stone mass. Maturity index (*MI*) was calculated according to the International Olive Council guidelines [18], where 100 fruits were randomly collected and scored from 0 to 7, according to the coloring stage of both skin and flesh, ranging from 0 as deep green skin color, to 7 as black skin color with all the flesh purple to the stone. Then, by applying Equation (1), where the number of fruits (from *A* to *H*) is estimated for each category (from 0 to 7), an *MI* value was obtained for each ripening stage.
(1)$$MI=\frac{A0+B1+C2+D3+E4+F5+G6+H7}{100}$$

Humidity (H) and fat content in fresh weight (FCFW) analyses were determined by NIR (near infra-red spectroscopy) technology (Bruker Optics, Madison, WI, USA), which has been demonstrated to be a very reliable and comparable technique for olive paste analysis [19].

### 2.4. Olive Paste Preparation and Laboratory Scale Olive Oil Extraction

Olive pastes were produced independently for each cultivar within the same day, right after fruit collection. Olives were crushed in a laboratory scale mill (ALREN, Spain) through a 5 mm pore grid. The obtained olive pastes were stored at −20 °C overnight, for subsequent olive oil extraction.

From the produced olive paste, portions of about 1 kg were weighed and the malaxation performed in a commercially available Yämmi 2 XL 1500 W food processor (Yämmi, Sonae MC, Matosinhos, Portugal), inside a 3.7 L stainless steel cylindrical-shaped vessel. This equipment allows selecting from 11 different mixing velocities, ranging from about 40 rpm, at velocity 1, to about 10,500 rpm, at velocity 11, respectively. According to the selected velocity, a centrally located mixing blade will be activated, and aided by a spatula mixer, will perform the malaxation process. This malaxation process was conducted without heating, at approximately 100 rpm, for 45 min. After this, paste was transferred to 50 mL Falcon tubes and centrifuged (Hermle Z 323 K, Gosheim, Germany) at 9000 rpm for 10 min. The oil fraction was then extracted, measured for yield calculation and stored in 50 mL dark glass bottles at −20 °C until analysis.

### 2.5. OO Yield Production Measurement

After centrifugation and oil extraction, the total OO volume was measured before storage, and the yield calculated as the ratio between the oil volume and the olive fruit weight from which it was obtained, considering the measured density of the obtained olive oils as 0.91 g/mL.

### 2.6. Industrial Scale Olive Oil Extraction

In parallel to the laboratory scale (LS) OO extraction, industrial scale (IS) extraction was performed with the two-phase olive mill from Herdade do Esporão. The process consisted in fruit crushing by a hammer mill to produce olive paste, followed by 30 min malaxation time at 25 °C in vertical malaxers. OO separation was obtained by centrifugation without water addition. IS OO were produced from the same orchards and within the same harvesting day as the LS.

### 2.7. FT-NIR Spectroscopy

A Bruker Optics FT-NIR spectrometer (Bruker Optics, Madison, WI, USA) was used for OO free acidity measurements, from both LS and IS OO extractions. All test samples were measured at a constant temperature of 50 °C in the transmission mode using 8 mm outer diameter glass disposable tubes (Bruker Optics, Madison, WI, USA). Instrument control and data processing were performed using OPUS v. 7.0 software.

### 2.8. K232, K268 and ΔK Measurements

Specific coefficients of extinction at 232 and 268 nm (K232 and K268) were evaluated according to the European Union Standard Methods [20]. About 0.05 g and 0.25 g of OO sample were weighed in 25 mL volumetric flasks, for K232 and K268, respectively, and diluted in iso-octane. Measurements were performed in triplicate, for each wavelength, in a Thermo Scientific (Waltham, MA, USA) Helios Beta spectrophotometer. Results were obtained using Equation (2):(2)$$K\lambda =\frac{E\lambda }{c\cdot s}$$
where *Kλ* is the specific extinction for wavelength *λ*, *Eλ* is the measured extinction on the wavelength *λ*, *c* is the solution concentration in g/100 mL and *s* is the thickness of measurement cell wall in cm. Specific extinction variation (Δ*K*) was measured according to Equation (3):(3)$$\Delta K=Km-\left(\frac{K_{m-4}+K_{m+4}}{2}\right)$$
where *Km* represents the specific extinction in the 268 nm wavelength.

### 2.9. Hydrophilic Phenolic Extracts

For the hydrophilic extraction of OO, approximately 10.00 ± 0.20 g of OO was weighed, and 10 mL of hexane followed by 20 mL of MeOH were added. The mixture was then agitated in a vortex for 1 min and phase separation performed by centrifugation (Hermle Z 323 K, Gosheim, Germany) for 10 min at 8000 rpm. Methanolic fraction was collected, and lipophilic fraction re-extracted twice, following the same process. The hydrophilic extract was then evaporated to dryness in a rotary evaporator (Heidolph Instruments, Schwabach, Germany) under low pressure at 35 °C. The final extract was dissolved in 2.0 mL of methanol and filtered through a polytetrafluoroethylene (PTFE) 0.22 µm syringe filter and stored at −20 °C before analysis. Triplicates were performed in three independent extractions.

### 2.10. Total Phenolic Compounds

Total phenolic compounds were determined by the Folin–Ciocalteu assay, by the adaptation of the Falleh et al. method [21]. In short, 150 μL of 10% (*v/v*) Folin–Ciocalteu reagent solution was added to an aliquot of 10 μL of the hydrophilic phenolic extract. The mixture was stirred and allowed to rest in the dark for 5 min. Then, 150 μL of a 60 g/L Na_2_CO_3_ solution was added, the mixture was stirred again and rested in the dark for 60 min, before reading at 725 nm on a microplate reader (MultiScan FC, Thermo Scientific, Waltham, MA, USA). A gallic acid calibration curve was prepared, ranging from 0.040 to 0.400 mg/mL, with results expressed as mg of GAE (gallic acid equivalent) per mL of extract.

### 2.11. GC-FID Analysis of Fatty Acids

Analysis of fatty acids in OO samples was performed through transesterification with cold methanol solution of KOH according to the official method [22]. Briefly, in glass tubes (of 5 mL), approximately 0.1 g of the olive oil sample was mixed with *n*-heptane (1:20, *w:v*) and, later, 0.2 mL of 2N solution of KOH in methanol was added. The final solution was agitated in a vortex for 30 s, and when the upper layer, containing fatty acid methyl esters (FAME), became clear, 1 mL of this solution was directly transferred to the vial for gas chromatography (GC) analysis.

FAME composition and quantification was performed by gas chromatography using a Hewlett Packard (6890 series) system equipped with a flame ionization detector (GC-FID). The analysis was performed using a Supelco, SP 2380 fused silica capillary column (60 m × 0.25 mm × 0.20 μm) (Bellefonte, PA, USA). The working conditions of the GC-FID were as follows: the injector and detector temperatures were set at 250 and 260 °C, respectively. 1 μL of sample was injected by the auto-sampler (Agilent, 7683) in split mode (20:1 ratio), the oven initial temperature was set at 140 °C, maintained for 5 min, and increased at 4 °C min^−1^ up to 240 °C and held for 10 min. The flow rate was set at 1.2 mL min^−1^ and the carrier gas was hydrogen. The total runtime of the analysis was 40 min. Advanced Chromatography Data Station—Clarity Software Solutions v. 7.4, was used for data acquisition, processing and instrument control.

The identification of fatty acids was performed by comparing retention times with the FAME-mix standard [23].

For quantification purposes, the % of each fatty acid area in relation to the total area of all fatty acids was calculated.

### 2.12. Data Analysis

For the statistical analyses of the experimental data, analysis of variance (ANOVA) was applied with Fisher test, for a confidence level of 95%. Principal component analysis (PCA) was also performed for GC-FID data. All analyses were performed using XLSTAT software (version 2022.4.1).

## 3. Results and Discussion

Three different cultivars were used to produce monovarietal OO, GV, COB (two traditional Portuguese cultivars for OO production) and ARB (the most predominant exotic cultivar implemented in Portugal, mainly in the Alentejo region). Prior to OO production, some basic characterizations of the fruit pastes were performed to better evaluate our samples according to their physical composition (Table 1).

The observed differences in MI, among the three cultivars, is justified by the fact that all samples were collected on the same date. Since each cultivar has its specific ripening process [24,25], differences are expected in the MI among cultivars, as observed. GV is a well-known early-ripening Portuguese cultivar [26], which, as expected, presented the highest MI at collection time (MI 3.24), while ARB was shown to still be at a very early ripening stage (MI 0.98). COB also presented a much later ripening when compared to GV, reaching MI 2.25 at harvest. Previous studies have reported different ripening evolutions for these cultivars, also showing GV as the earliest ripening cultivar [27]. FW, SW, FP and FP/S were observed to be within a much higher range for COB, with 4.43 ± 1.06 g, 0.60 ± 0.12 g, 3.84 ± 0.84 g and 6.39 ± 0.42, respectively, which was expected since COB is a well-known traditional Portuguese cultivar, known for its high fruit caliber [27]. On the other hand, ARB presented the lowest FW, SW, FP and FP/S, with values of 1.28 ± 0.24 g, 0.25 ± 0.07 g, 1.03 ± 0.16 g and 4.32 ± 0.93, respectively, which was also expected, since ARB is characterized by its small fruit weight and general reduced tree architecture [28], for this reason being one of the most well adapted cultivars for high density hedgerow orchards [29]. Furthermore, its high oil yields are recognized and well appreciated [30,31], which corroborates with our results, that show ARB as the cultivar with highest fat content in fresh weight (FCFW), with values of 14.96 ± 0.41%, significantly higher (*p*-value < 0.05) than that obtained for other cultivars, where GV and COB showed FCFW values of 13.40 ± 0.37% and 14.11 ± 0.05%, respectively. For all cultivars, high levels of moisture were observed, with humidity levels ranging from 58.52 ± 0.54% to 61.15 ± 0.16% in ARB and GV, respectively, while COB registered a humidity level of 60.40 ± 1.17%. Olive fruit moisture may vary significantly depending on several factors, such as meteorological conditions, agronomic practices (i.e., orchard irrigation) as well olive fruit ripening stage [32]. Olive paste moisture is directly related to the OO extraction efficiency, with pastes having humidity levels higher than 50% classified as “difficult pastes”, with OO extraction yields decreasing with higher moisture percentages [33]. Therefore, ARB samples, presenting significantly higher levels of FCFW and lower moisture content, should be seen as the best samples for higher OO extraction yield. Therefore, in order to assess OO production yields for the three cultivars under study, calculations were performed with the implemented laboratory OO production method, with results shown in Table 2.

As observed, and in accordance with FCFW measurements, highest OO production yield was obtained for the ARB cultivar, with about 8%, while 7.6% was measured for COB and 6.6% for GV, respectively. With an MI increase, an FCFW increase is also expected, and therefore, higher extraction yields, but, as shown by Ferro et al. [27] for GV and COB, despite its high correlation, this relation is cultivar specific, thus maximum FCFW accumulation may be obtained at different MI, depending on the cultivar. As shown, ARB presented higher FCFW accumulation at much lower MI, leading to higher extraction yields. Compared with other OO extraction systems, such as the well-known and industrially implanted Abencor [16], for the ARB cultivar, Franco et al. [32] showed an OO extraction yield for an MI classified as “green” (<2) of near 11%, about 3% higher than that obtained with our laboratory extraction method. The observed extraction yield difference may be explained by the difference in ripening indexes, since in our work the ARB cultivar was processed at a very early ripening stage, with an MI lower than 1 (deep green coloration), in contrast to the “green” (MI < 2) classification shown in the compared Abencor work. Thus, within cultivar, higher MI may lead to a OO extraction yield increase, as also shown by Franco et al. [32], since the fat content tends to increase to its maximum accumulation, as shown by Ferro et al. [27]. Thus, the 3% difference observed in the extraction yields of both methodologies may be due to the fact that different ripening stages were considered. Furthermore, in our study, we obtained FCFW values of 14.96 ± 0.41% for ARB (Table 1), while Franco et al. [32] showed an FCFW of 17.6 ± 1.4% for “green” ARB, being the increased FCFW content also a major contributor for higher OO extraction yields. In addition, samples from our study presented a considerably higher moisture content, 58.52 ± 0.54%, compared to the 53.8 ± 2.7% obtained by Franco et al. [32], which, as previously mentioned, difficult the OO extraction, and, consequently, its extractability and extraction yield.

Extractability of olive paste was also measured, in order to assess the performance of our implemented technology in recovering the potential OO present in the paste. Extractability is the percentage of OO extracted from the total fruit oil content (on a fresh matter basis). Thus, extractability was calculated between the maximum potential of OO present in the olive paste, measured by FCFW, and the real OO extracted by the extraction method, assessed by OO yield. Obtained results are expressed in Figure 1 and showed highest extractability for COB and ARB, with about 53%, while GAL showed only 50% extractability. The lower percentage observed for GAL may be related to the high moisture content observed in this cultivar (>61%). High moisture leads to the formation of larger and stronger emulsions during the milling phase, which are impossible to break under the applied malaxation conditions, mainly without temperature addition [34].

The obtained extractability ranged from 50 to 53%, which may be seen as relatively low percentages when compared with studies from other authors with higher extractability values [32,33,35,36]. It is important to mention that the proposed laboratory extraction method does not use any extraction additive to enhance OO availability. It is well described that those additives, p.e. microtalc, might be added to “difficult pastes” to adsorb the natural emulsifiers from the surface of the OO droplets, increasing the extracted oil as well its extractability [32,33,37]. In our study we decided not to add any type of extraction enhancer, to correctly evaluate the performance of the developed method, therefore lower extractability percentages were expected.

In order to further assess the behavior of our extraction method in regard to OO quality, olive samples from the same orchard and collected on the same date, were processed into olive pastes for OO extraction at the industrial scale (IS) and in parallel at the laboratory scale (LS). Produced OO, by both extraction procedures, were analyzed in terms of basic chemical composition (Table 3).

All the produced OO, both from the IS and LS extraction, showed values below the maximum regulated limits for the physicochemical parameters (free acidity, K232, K268 and ΔK), classifying them into the “extra virgin” category according to the Commission Regulation [20]. The obtained results underly the relevance of the proposed LS method for OO extraction. The comparison between LS and IS OO, for each cultivar, reveals differences (*p*-value < 0.05) for almost all variables, demonstrating that in fact, significant differences exist between the two OO extraction methods. As for free acidity, no significant differences (*p*-value ≥ 0.05) were observed between GV samples, regardless of the extraction procedure, while ARB showed a lowered acidity for OO LS produced, and opposite results were observed for COB samples. Despite the differences in ARB and COB, OO samples were within the same order of magnitude, which reveal a clear cultivar chemical distinction. The results obtained from the LS OO extraction revealed lower K232 and K268 values than the ones obtained from the IS, which may occur since olive samples for the LS method were all hand-picked and processed right after collection (on the same day), thus not subjected to any mechanical degradation factor, which may occur at the industrial level. Furthermore, other factors within the olive fruit processing chain could account for these deviations, namely, the time between olive collection and OO extraction, fruit degradation due to the mechanical collection, fruit transportation and storage. Total phenol measurement showed significantly higher levels for GV and ARB OO obtained at IS, revealing, for these two cultivars, the effectiveness of IS extraction to recover phenolic compounds. For COB no significant differences were detected between both extraction processes. Despite the observed differences for GV and ARB, the strong cultivar effect on both extraction methods was once more quite noticeable, with COB showing generally higher values for total phenolic compounds.

In order to further assess the quality parameter of the produced OO, the fatty acid composition was analyzed with results shown in Table 4. A total of 21 fatty acids were able to be identified and their relative proportions quantified in all samples, with the exception of erucic acid (C22:1n9), which was not identified in ARB samples. As expected, oleic acid (C18:1) was the main fatty acid found in all cultivars for both extraction methods, ranging from 66.38 ± 0.17% and 67.979 ± 0.026%, for ARB LS and IS, respectively, to 71.59 ± 0.12% and 72.094 ± 0.089%, for GV LS and IS, respectively. For all cultivars, significantly higher (for a *p*-value < 0.05) percentages of oleic acid were observed with IS OO extraction, despite the lower dispersion between both extraction methods, ranging from 2.08% for COB to 0.50% for GV, respectively. In contrast, palmitic acid (C16:0), as the second major fatty acid, presented significantly higher (for a *p*-value < 0.05) percentages for the LS OO extraction, with a dispersion from the IS ranging from 1.18% for GV to 0.64% for COB, respectively. Linoleic acid (C18:2) was shown to be highly cultivar specific, with higher percentages for ARB, ranging from 10.867 ± 0.072% and 10.1749 ± 0.0018%, for ARB LS and IS, respectively, to 4.233 ± 0.013% and 4.3825 ± 0.0070%, for GV LS and IS, respectively. As shown, these differences on major fatty acids are well related with cultivar specificity, rather than the OO extraction method. In agreement, total SFA was shown to be highest for the ARB cultivar with LS extraction, with 20.03 ± 0.26%, while with IS extraction only 18.885 ± 0.043% were measured. On the other hand, GV showed a more similar SFA measurement for both extraction methods, with no significant difference between them (for a *p*-value ≥ 0.05), with 19.26 ± 0.13% and 19.095 ± 0.09% for LS and IS, respectively, while COB ranged from 19.06 ± 0.12% to 18.298 ± 0.056% for LS and IS, respectively. ARB also presented lowest MUFA measurements for both extractions (68.79 ± 0.18% and 70.271 ± 0.034%, for LS and IS, respectively) and highest PUFA values (11.178 ± 0.072% and 10.848 ± 0.010%, for LS and IS, respectively), which was also expected since ARB showed the lowest oleic acid and highest linoleic acid, major MUFA and PUFA fatty acids, respectively. On the contrary, GV presented the highest MUFA (75.36 ± 0.11% and 75.245 ± 0.071%, for LS and IS, respectively) and lowest PUFA values (5.359 ± 0.034% and 5.658 ± 0.026%, for LS and IS, respectively). Furthermore, all samples from both extraction methods were compared to the European Community Regulation for EVOO [20] and, regarding the fatty acid profile, all fell within established and regulated levels.

Fatty acid composition of OO is well known to be cultivar-related, but with a great contribution by the edaphoclimatic growth conditions [30,38]. In the present study, these variability factors were reduced to a minimum since all samples were collected from the same orchard, therefore, the observed fatty acid variations can be mainly accounted for both cultivar and OO extraction methodology. Extraction conditions are placed among the major variability factors for OO quality, with temperature as a critical condition. Hilali et al. [39] showed that different extraction conditions regarding the use of temperature negatively influence the qualitative parameter of OO, namely, acidity and peroxide index. Regarding the fatty acid profile, the author also observed a stearic acid (C18:0) and linoleic acid (C18:2) reduction, due to the heat increment in the extraction. Therefore, the applied LS OO extraction conditions were kept as similar to the ones used in the industry as possible, in order to reduce variability to its minimum.

For a more comprehensive visualization of the analyzed fatty acids from different cultivar obtained by the two extracting systems, exploratory analysis was performed by PCA, with observations plotted in Figure 2.

As one can see, three major clusters were formed among the observations, showing a clear correlation of the different extraction systems for the same cultivar, with about 78% explained variability for the two first principal components (F1 and F2). These results reveal that despite the two different extraction systems applied, the fatty acid proportion on each sample remained similar enough to enable a distinct cultivar separation.

## 4. Conclusions

With this work we have shown an alternative laboratory scale method for OO extraction, accessible for any analytical laboratory to implement, showing promising results in terms of extraction yields and OO extractability, as well as regarding its quality parameters. Among the three tested cultivars, a clear distinction of the cultivar variability was observed regardless of the extraction method applied. Despite the significant differences observed in some of the tested parameters, namely, K232, K268 and total phenols, all OO produced could still be classified as extra virgin olive oil (within the evaluated parameters). Nevertheless, further validation of the proposed method should be performed by applying a direct comparative evaluation with OO extracted with the Abencor system.

## Figures and Tables

**Figure 1 foods-12-00380-f001:**
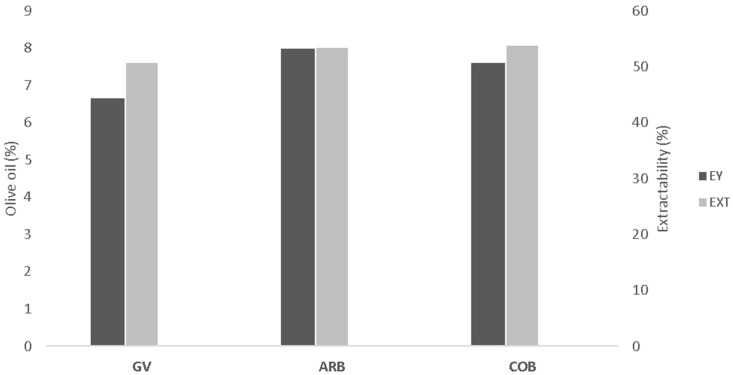
Olive oil extraction yield (EY, dark grey) to be read at the left axis. Olive oil extractability (EXT, bright grey), to be read at the right axis. Olive cultivars: ‘Galega vulgar’ (GV), ‘Arbequina’ (ARB) and ‘Cobrançosa’ (COB).

**Figure 2 foods-12-00380-f002:**
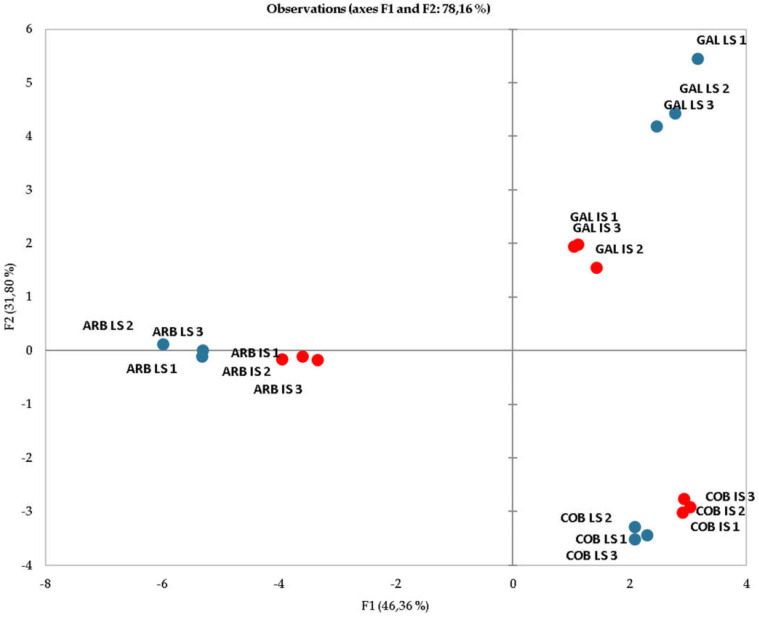
Principal component analysis (PCA) of the fatty acid characterization of olive oils (OO) obtained by two different extraction methods, laboratory scale (LS) and industrial scale (IS), analyzed in triplicate for three cultivars: GV ‘Galega vulgar’, COB ‘Cobrançosa’ and ARB ‘Arbequina’.

**Table 1 foods-12-00380-t001:** Measurement of maturity index (MI), fruit weight (FW), stone weight (SW), fruit pulp (FP), fruit pulp to stone ratio (FP/S), fat content in fresh weight (FCFW) and humidity (H), for three olive cultivars—‘Galega vulgar’ (GV), ‘Arbequina’ (ARB) and ‘Cobrançosa’ (COB). The values represent the measurement of three assays and the associated standard deviation.

Cultivar	MI	FW (g)	SW (g)	FP (g)	FP/S	FCFW (%)	H (%)
GV	3.24	2.56 ± 0.41 ^b^	0.38 ± 0.08 ^b^	2.19 ± 0.33 ^b^	5.90 ± 0.49 ^a,b^	13.40 ± 0.37 ^c^	61.15 ± 0.16 ^a^
ARB	0.98	1.28 ± 0.24 ^c^	0.25 ± 0.07 ^c^	1.03 ± 0.16 ^c^	4.32 ± 0.93 ^b^	14.96 ± 0.41 ^a^	58.52 ± 0.54 ^b^
COB	2.25	4.43 ± 1.06 ^a^	0.60 ± 0.12 ^a^	3.84 ± 0.84 ^a^	6.39 ± 0.42 ^a^	14.11 ± 0.05 ^b^	60.40 ± 1.17 ^a,b^

^a–c^: Different superscripts differ significantly (*p*-value < 0.05).

**Table 2 foods-12-00380-t002:** Olive oil (OO) laboratory scale (LS) production yield for three olive cultivars: ‘Galega vulgar’ (GV), ‘Arbequina’ (ARB), ‘Cobrançosa’ (COB), with reference to the produced olive paste (OP).

Cultivar	OP (g)	OO (g)	Yield (%)
GV	4379.4	291.2	6.65
ARB	3076.3	245.7	7.99
COB	4434.8	336.7	7.59

**Table 3 foods-12-00380-t003:** Basic quality characterization (free acidity (%), K232, K268, ΔK and total phenols (mg EAG/kg)) of monovarietal olive oils (OO) from the ‘Galega vulgar’ (GV), ‘Arbequina’ (ARB) and ‘Cobrançosa’ (COB) cultivars, produced by laboratory scale (LS) and industrial scale (IS) extraction methods. Results are presented by the mean value of three independent measurements ± standard deviation.

Cultivar	Extraction Method	Free Acidity	K232	K268	ΔK	Total Phenols
GV	LS IS	0.082 ± 0.017 ^a^ 0.096 ± 0.022 ^a^	1.428 ± 0.002 ^b^ 1.742 ± 0.013 ^a^	0.081 ± 0.002 ^b^ 0.152 ± 0.007 ^a^	−0.0094 ± 0.0003 ^b^ −0.001 ± 0.004 ^a^	328.2 ± 2.5 ^b^ 493.1 ± 17.3 ^a^
ARB	LS IS	0.038 ± 0.005 ^b^ 0.075 ± 0.021 ^a^	1.572 ± 0.007 ^b^ 1.856 ± 0.004 ^a^	0.141 ± 0.003 ^b^ 0.204 ± 0.005 ^a^	−0.009 ± 0.004 ^b^ 0.008 ± 0.002 ^a^	248.3 ± 16.6 ^b^ 350.0 ± 28.7 ^a^
COB	LS IS	0.147 ± 0.013 ^a^ 0.112 ± 0.012 ^b^	1.710 ± 0.009 ^b^ 1.833 ± 0.006 ^a^	0.176 ± 0.002 ^b^ 0.185 ± 0.005 ^a^	0.0005 ± 0.0034 ^a^ −0.009 ± 0.007 ^a^	688.6 ± 28.9 ^a^ 690.3 ± 30.8 ^a^

^a,b^: Different superscripts differ significantly, at a 5% significance level (*p*-value < 0.05), between LS and IS results for each cultivar, according to ANOVA results applying the Fisher test.

**Table 4 foods-12-00380-t004:** Fatty acid composition (%) and total saturated fatty acids (SFA), monounsaturated fatty acids (MUFA) and polyunsaturated fatty acids (PUFA) of olive oils (OO) extracted from both methods, the laboratory scale (LS) and the industrial scale (IS), for the three studied cultivars: ‘Arbequina’ (ARB), ‘Cobrançosa’ (COB) and ‘Galega vulgar’ (GV). Measurements were performed in triplicate and results expressed by mean ± standard deviation. Limit of detection (LOD) was indicated for compounds identified below the detection limit.

Name	Abbreviation	ARB		COB		GV	
		LS	IS	LS	IS	LS	IS
Myristoleic acid	C14:0	0.0189 ± 0.0016 ^a^	0.01645 ± 0.00027 ^b^	0.00809 ± 0.00063 ^a^	0.00898 ± 0.00039 ^a^	0.0109 ± 0.0012 ^a^	0.01167 ± 0.00024 ^a^
Pentadecanoic acid	C15:0	0.01363 ± 0.00041 ^a^	0.01319 ± 0.00039 ^a^	0.0071 ± 0.0011 ^a^	0.00703 ± 0.00011 ^a^	0.00976 ± 0.00051 ^b^	0.01279 ± 0.00027 ^a^
Palmitic acid	C16:0	16.98 ± 0.25 ^a^	15.885 ± 0.060 ^b^	15.07 ± 0.11 ^a^	14.427 ± 0.052 ^b^	17.07 ± 0.14 ^a^	15.89 ± 0.10 ^b^
Palmitoleic acid	C16:1	1.3522 ± 0.0066 ^a^	1.2548 ± 0.0023 ^b^	1.2548 ± 0.0063 ^a^	1.1149 ± 0.0061 ^b^	2.644 ± 0.021 ^a^	2.1397 ± 0.0096 ^b^
Heptadecanoic acid	C17:0	0.1776 ± 0.0013 ^a^	0.1781 ± 0.0023 ^a^	0.1269 ± 0.0013 ^b^	0.13208 ± 0.00070 ^a^	0.11091 ± 0.00092 ^b^	0.16037 ± 0.00084 ^a^
cis-10-heptadecenoic acid	C17:1	0.3392 ± 0.0016 ^a^	0.333 ± 0.010 ^a^	0.21730 ± 0.00067 ^b^	0.22925 ± 0.00032 ^a^	0.3101 ± 0.0047 ^a^	0.322 ± 0.016 ^a^
Stearic acid	C18:0	2.118 ± 0.011 ^a^	2.134 ± 0.014 ^a^	3.215 ± 0.019 ^a^	3.0968 ± 0.0050 ^b^	1.570 ± 0.012 ^b^	2.3958 ± 0.0083 ^a^
Oleic acid	C18:1n9c	66.38 ± 0.17 ^b^	67.979 ± 0.026 ^a^	68.757 ± 0.083 ^b^	70.842 ± 0.054 ^a^	71.59 ± 0.12 ^b^	72.094 ± 0.089 ^a^
Linolelaidic acid	C18:2n6t	0.0087 ± 0.0017 ^a^	0.00827 ± 0.00070 ^a^	0.00729 ± 0.00091 ^a^	0.00634 ± 0.00029 ^a^	0.00592 ± 0.00011 ^b^	0.00803 ± 0.00017 ^a^
Linoleic acid	C18:2n6c	10.867 ± 0.072 ^a^	10.1749 ± 0.0018 ^b^	8.709 ± 0.023 ^a^	7.5731 ± 0.0038 ^b^	4.233 ± 0.013 ^b^	4.3825 ± 0.0070 ^a^
Arachidic acid	C20:0	0.4340 ± 0.0021 ^a^	0.4363 ± 0.0027 ^a^	0.4371 ± 0.0014 ^a^	0.4282 ± 0.0017 ^b^	0.3141 ± 0.0072 ^b^	0.4124 ± 0.0031 ^a^
cis-11-Eicosenoic acid	C20:1	0.7566 ± 0.0063 ^a^	0.69019 ± 0.00090 ^b^	0.9368 ± 0.0023 ^a^	0.8717 ± 0.0011 ^b^	0.8006 ± 0.0028 ^a^	0.68235 ± 0.00073 ^b^
Linolenic acid	C18:3n3	0.2877 ± 0.0015 ^b^	0.2912 ± 0.0017 ^a^	0.2118 ± 0.0011 ^a^	0.2142 ± 0.0025 ^a^	0.2836 ± 0.0064 ^a^	0.2562 ± 0.0018 ^b^
Heneicosanoic acid	C21:0	0.01759 ± 0.00091 ^a^	0.0162 ± 0.0010 ^a^	0.01041 ± 0.00023 ^a^	0.01014 ± 0.00028 ^a^	0.01631 ± 0.00082 ^a^	0.01561 ± 0.00025 ^a^
cis-11,14-Eicosadienoic acid	C20:2	0.00282 ± 0.00012 ^a^	0.00245 ± 0.00024 ^a^	0.00139 ± 0.00091 ^b^	0.001921 ± 0.000061 ^a^	0.00204 ± 0.00040 ^a^	0.00141 ± 0.00025 ^b^
Behenic acid	C22:0	0.1366 ± 0.0027 ^a^	0.1352 ± 0.0036 ^a^	0.10964 ± 0.00044 ^b^	0.1151 ± 0.0015 ^a^	0.0997 ± 0.0039 ^b^	0.12404 ± 0.00047 ^a^
Erucic acid	C22:1n9	LOD	LOD	0.00199 ± 0.00012 ^a^	0.00204 ± 0.00011 ^a^	0.00255 ± 0.00021 ^a^	0.00257 ± 0.00048 ^a^
Arachidonic acid	C20:4n6	0.02619 ± 0.00061 ^b^	0.3628 ± 0.0059 ^a^	0.8308 ± 0.0092 ^a^	0.8405 ± 0.0037 ^b^	0.832 ± 0.026 ^b^	1.009 ± 0.018 ^a^
Lignoceric acid	C24:0	0.0717 ± 0.0013^a^	0.06972 ± 0.00088 ^a^	0.0731 ± 0.0016 ^a^	0.0717 ± 0.0023 ^b^	0.0558 ± 0.0038 ^b^	0.0676 ± 0.0033 ^a^
cis-5,8,11,14,17-Eicosapentaenoic acid	C20:5n3	0.00326 ± 0.00052 ^b^	0.00499 ± 0.00037 ^a^	0.0047 ± 0.0023 ^a^	0.00416 ± 0.00048 ^a^	0.025 ± 0.026 ^b^	0.0036 ± 0.0016 ^a^
Nervonic acid	C24:1	0.00490 ± 0.00031 ^b^	0.0110 ± 0.0015 ^a^	0.00502 ± 0.00034 ^a^	0.00165 ± 0.00036 ^b^	0.00358 ± 0.00022 ^a^	0.00250 ± 0.00028 ^b^
SFA		20.03 ± 0.26 ^a^	18.885 ± 0.043 ^b^	19.06 ± 0.12 ^a^	18.298 ± 0.056 ^b^	19.26 ± 0.13 ^a^	19.095 ± 0.094 ^a^
MUFA		68.79 ± 0.18 ^b^	70.271 ± 0.034 ^a^	71.173 ± 0.083 ^b^	73.062 ± 0.047 ^a^	75.36 ± 0.11 ^a^	75.245 ± 0.071 ^a^
PUFA		11.178 ± 0.072 ^a^	10.848 ± 0.010 ^b^	9.764 ± 0.032 ^a^	8.636 ± 0.010 ^b^	5.359 ± 0.034 ^b^	5.658 ± 0.026 ^a^

^a,b^: Different superscripts differ significantly, at a 5% significance level (*p*-value < 0.05), between LS and IS results for each cultivar, according to ANOVA results applying the Fisher test.

## Data Availability

The data that support the findings of this study are available from the corresponding author, Duarte MF, upon reasonable request.

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
