# Peer review of "A New Laboratory Scale Olive Oil Extraction Method with Comparative Characterization of Phenolic and Fatty Acid Composition"

_foods, 2023, doi:10.3390/foods12020380_

Round 1

Reviewer 1 Report

The manuscript is interesting and showed good goals to understand the different extraction process of olive oil, mainly, at non-industrial scale.

Line 40-41: in the sentence “oil accumulation starts to occur by the beginning of July (5 to 7 weeks after full bloom) and finishes by the end of October” authors must specify whether the date refers to the southern or northern hemisphere

Line 141-143: the author can provide details about the industrial method of olive oil extraction used

Line 145: in sentences “spectrometers Bruker …” and “disposable tube Bruker …” reduce the space between words

Line 199: separate the unit from the numeral

Line 223-230: the different maturation index obtained from the three different cultivars can interfere in extraction process? Please the authors can provide some discussion about this effect on oil extraction method either industrial or non-industrial process. Some discussions are showed in line 262-280, but is important correlate the MI to facility or difficult the olive paste production and extraction methos or olive oil yield

Line 194: In Figure 1, please the author can show the error bars for all cultivars and parameters evaluated. The bars are not good, please the author would put the bars side by side, not as it is, as it is difficult to visualize. In the Y right axis (Extractability), please use the real scale (0 to 55%), not as presented.

Line 397: “two first principal components (F1 and F2).”, the authors can provide the parameters/fatty acids allocated in each component F1 and F2 in PCA analysis.

Line 400-407: the authors affirm that the analyzed method is similar when compared with the industrial method, however, in many parameters evaluated, the methods when compared showed differences, mainly in phenolic compounds and other olive oil parameters. The authors can review and improve this part of manuscript.

Author Response

Please see the rebuttal letter in attachment.

Reviewer 2 Report

Dear authors,

The manuscript is very interesting in terms of providing a comparison of olive oil extraction in a laboratory environment and in the industrial sector. The subject is very well laid out in the introduction, and the materials and methods are clearly described. The results should be presented in accordance with the journal guidelines. This specifically refers to tables and references. The manuscript should also include section 4 Conclusion. Provided the above changes are made:

Proposed changes in specific lines:

Line:

28 The numbers in the abstract should be presented without the deviation from the mean value.

32 Key words should be listed in the alphabetical order. Try finding key words that are not already included in the manuscript title.

102 Add “.” at the end of the sentence.

212 It is sufficient to provide the full name of the variety once, along with the abbreviated name. Only use the abbreviated name in the rest of the text.

218 Small letters pointing to the significance of the difference should be used in such a way that “a” signifies the highest value (e.g. 2.56 ± 0.41b /1.28 ± 0.24c / 4.43 ± 1.06a)

311 Edit the title of Table 3 (Basic quality characterization (Free Acidity (%) K232 K268 ΔK) and Total phenols ((mg EAG/Kg) of monovarietal olive…)

383 Table 4 is not entirely visible

400 Add the header “4 Conclusion” to the section

References:

General: omit the paper DOI

427 Incomplete source details

498 Correctly quote the names of the authors (Jimenez, A. Kneading)

503 Omit “2011”

510 provide page numbers

512 quote all of the authors and the journal pages

Author Response

(The authors gave the same response as above.)

Reviewer 3 Report

The manuscript describes an alternative to olive oil extraction, evaluating the extractability, physicochemical parameters, phenolic content and fatty acid profile. For a better understanding, some points must be taken into consideration.

It is not clear what the contribution is, nor the problem that is being addressed. Manuscript presents a high similarity index (31 %), for that, work must be done on the document to reduce this value.

Line 60: Replace “minutes” by “min”

Line 65: “The optimization of OO extraction procedure enhances the activity of enzymes...” Which enzymes?

Methodology

Section 2.6. What is industrial extraction? Same laboratory conditions at higher volume? Was the sample processing the same?

Line 181: Indicate the range of calibration curve.

Authors indicate that three replicates were performed, but no standard deviations are reported in the results (Table 2, Fig. 1).

How was the experiment set up, under what experimental design? What were the factors and the independent variable, and what is the statistical analysis?

Fig.1: Values of FCFW already showed in table 1, not to be repeated. Likewise, add the error bars and significance letters.

Line 371: “Fatty acid composition of OO is well known to be cultivar-related…”. What is the main contribution of the study?

Line 376: “Extraction conditions are placed among the major variability factor on OO quality, with temperature as a critical condition”. If temperature is a critical factor, why was it not considered in the study?

Table 4: Statistical analysis is needed to stablish differences between extraction method and/or olive fruit variety.

 What are your conclusions? add a section.

Author Response

(The authors gave the same response as above.)
